# Friendship Bench intervention to address depression and improve HIV care engagement among adolescents living with HIV in Malawi: Study protocol for a pilot randomized controlled trial

**Thuy Thi Dieu Dao**[1,2][*], **Bradley N. Gaynes**[1,3], **Brian W. Pence**[1], **Steven M. Mphonda**[4], **Kazione Kulisewa**[5], **Michael Udedi**[6], **Melissa A. Stockton**[7], **Jack Kramer**[5], **Katherine Grace Waddell**[4], **Maria Faidas**[1], **Hillary Mortensen**[1], **Nivedita L. Bhushan**[8]

1 Department of Epidemiology, Gillings School of Global Public Health, University of North Carolina at Chapel Hill, Chapel Hill, North Carolina, USA, 2 Center for Research and Training on Substance Abuse and HIV, Hanoi Medical University, Hanoi, Vietnam, 3 Department of Psychiatry, University of North Carolina at Chapel Hill, Chapel Hill, North Carolina, USA, 4 University of North Carolina Project Malawi, Lilongwe, Malawi, 5 Department of Mental Health, Kamuzu University of Health Sciences, Blantyre, Malawi, 6 NCDs & Mental Health Division, Malawi Ministry of Health, Lilongwe, Malawi, 7 Department of Psychiatry, University of Pennsylvania, Philadelphia, USA, 8 Center for Communication and Engagement Research, RTI International, Research Triangle Park, North Carolina, USA

* thuydao@unc.edu

## Abstract

### Background

Adolescents in Sub-Saharan Africa are disproportionately affected by the HIV epidemic. Comorbid depression is prevalent among adolescents living with HIV (ALWH) and poses numerous challenges to HIV care engagement and retainment. We present a pilot trial designed to investigate feasibility, fidelity, and acceptability of an <u>a</u>dapted and an <u>e</u>nhanced <u>F</u>riendship <u>B</u>ench intervention (henceforth: AFB and EFB) in reducing depression and improving engagement in HIV care among ALWH in Malawi.

### Methods

Design: Participants will be randomized to one of three conditions: the Friendship Bench intervention adapted for ALWH (AFB, n = 35), the Friendship Bench intervention enhanced with peer support (EFB, n = 35), or standard of care (SOC, n = 35). Recruitment is planned for early 2024 in four clinics in Malawi.

Participants: Eligibility criteria (1) aged 13–19; (2) diagnosed with HIV (vertically or horizontally); (3) scored ≥ 13 on the self-reported Beck's Depression Inventory (BDI-II); (4) living in the clinic's catchment area with intention to remain for at least 1 year; and (5) willing to provide informed consent.

Interventions: AFB includes 6 counseling sessions facilitated by young, trained nonprofessional counselors. EFB consists of AFB plus integration of peer support group sessions to facilitate engagement in HIV care. SOC for mental health in public facilities

---

**Data availability statement:** No datasets were generated or analyzed during the current study as a study protocol article. All relevant data from this study will be made available upon study completion and request.

**Funding:** This study was funded by the National Institute of Mental Health, grant number R34 MH130232 awarded to B.N.G and N.L.B. The funder did not and will not play any role in the study design, data collection and analysis, decision to publish, or preparation of the manuscript.

**Competing interests:** The authors have declared that no competing interests exist.

in Malawi includes options for basic supportive counseling, medication, referral to mental health clinics or psychiatric units at tertiary care hospitals for more severe cases. Outcomes: The primary outcomes are feasibility, acceptability, and fidelity of the AFB and EFB assessed at 6 months and 12 months and compared across 3 arms. The secondary outcome is to assess preliminary effectiveness of the interventions in reducing depressive symptoms and improving HIV viral suppression at 6 months and 12 months.

## Discussion

This pilot study will provide insights into youth-friendly adaptations of the Friendship Bench model for ALWH in Malawi and the value of adding group peer support for HIV care engagement. The information gathered in this study will lead to a R01 application to test our adapted intervention in a large-scale cluster randomized controlled trial to improve depression and engagement in HIV care among ALWH.

## Trial registration

ClinicalTrials.gov (NCT06173544)

## Introduction

Adolescents living in Sub-Saharan Africa (SSA) are disproportionately affected by the HIV epidemic. In 2022, SSA adolescents accounted for approximately 79% (1.3 million) of the estimated 1.65 million adolescents living with HIV (ALWH), aged 10–19, worldwide [1]. ALWH are vulnerable to mental health problems: ALWH are both living with a chronic disease and transitioning through a unique and formative period which includes intense physical, emotional and social changes [2,3]. Indeed, comorbid depression is prevalent and burdensome amongst ALWH in SSA with estimates of depressive symptoms among ALWH ranging from 12% to 45% [4]. Depression is a significant threat to engagement and retention in HIV care as well as worsening HIV-related outcomes among those receiving HIV treatment [5–9]. Considering the dual burden of HIV and depression, developing timely, developmentally appropriate, and effective interventions for ALWH is critical [4].

Pharmacological treatments are challenging due to cost and availability of designated professionals and are often reserved for treating moderate to severe cases of depression. In contrast, psychological interventions are preferable and widely implemented for adolescents and young adults with or without HIV in resource-limited countries [10–12]. Psychological interventions for depression comprise of various evidence-based approaches such as cognitive behavioral therapy (CBT) [11,13], interpersonal therapy (IPT) [14], motivational interviewing, family strengthening interventions [15], and problem-solving therapy [11]. While there is evidence that these interventions are effective among adults living with HIV in SSA, few studies have reported on their overall effectiveness for ALWH or the effectiveness of specific intervention components [16,17].

In Malawi, Teen Clubs are the predominant method of providing support for adherence to antiretroviral therapy (ART) and psychosocial support for ALWH. Teen Clubs are facilitated monthly by the Malawi Ministry of Health (MOH) clinician and trained lay workers in MOH clinic space to exclusively provide support for ALWH aged 10–19 years (size of a teen club ranges from 15–200 ALWH) [18]. During Teen Clubs, ALWH can pick up medication, interact with each other, receive sexual and reproductive health education, and obtain support for status disclosure and positive living [18]. At some clinics, psychosocial counselors are also

occasionally available to provide mental health counseling for ALWH [19]. Other programs in the country have utilized community engagement and mental health literacy approaches to improve psychosocial support for adolescents and young adults but these programs were not HIV specific [20–24]. Take together, there is a clear need for interventions which combine psychotherapy for addressing depressive symptoms and peer support for HIV care engagement for ALWH.

Task-sharing psychological interventions between specialist and non-specialist health providers has been indicated as a promising approach to address this treatment gap [25,26]. Specifically, counseling interventions delivered by non-specialists like lay health workers in resource-limited settings can alleviate depressive symptoms [27] and peer support can improve engagement in HIV care [28]. For adolescents, few data demonstrate the feasibility, acceptability, and effectiveness of such mental health interventions in combination with peer support to improve ALWH engagement in HIV care [29]. One promising intervention is the Friendship Bench (FB), an evidence-based problem-solving therapy intervention delivered by lay health workers that decreases depression among adults [30,31]. The original Friendship Bench consists of six individual counseling sessions plus optional group peer support [31]. It is currently being adapted for adolescents in Botswana but has not been specifically adapted for ALWH [32]. Although the FB intervention has been delivered to ALWH in Zimbabwe, only qualitative results have been published so far and these do not specifically include peer support for HIV care engagement [33–35]. Accordingly, we are proposing to adapt FB for ALWH in Malawi and pilot a clinical trial to test whether we can effectively deliver FB. If successful, we will test the effectiveness in a fully powered trial.

This paper aims to present the protocol of a randomized clinical pilot trial designed to investigate feasibility, fidelity, acceptability of the adapted and enhanced Friendship Bench in reducing depression and improving engagement in HIV care among ALWH in Malawi. The secondary outcome of the pilot trial is to assess the preliminary effectiveness of the adapted and enhanced Friendship Bench in comparison with standard of care.

## Methods

### Overview of study design

We will conduct a three-arm individual-level clinical pilot trial across four clinics in Malawi. Participants will be individually randomized to the Friendship Bench intervention adapted for adolescents living with HIV called the Adapted Friendship Bench (AFB) arm (n = 35), the Friendship Bench intervention enhanced with peer support called the Enhanced Friendship Bench (EFB) arm (n = 35), or standard of care (SOC) arm (n = 35). The study team will generate a random sequence using computer-based software (the rand() function in Microsoft 365 Excel version 2410). Allocation concealment will be ensured, the randomization code will not be released until the participants have been enrolled in the trial. Blinding might not be possible to achieve due to the nature of intervention. Recruitment and follow-up are planned from the 1st of June 2024 to the 31st of May 2025. Participants will be followed up to 12 months after their participation (Fig 1).

### Study settings

This pilot trial will be implemented in four health centers which are located in urban Lilongwe, Malawi with ART integrated in the clinics including Area 18, Area 25, Kawale, and Lighthouse. These centers were selected based on their homogeneity regarding staffing levels, service offered, NGO/ancillary program involvement (e.g., all four centers receive training and supervision support from Lighthouse Trust NGO in Lilongwe), patient volume and patient

| | STUDY PERIOD | | | | | |
|---|---|---|---|---|---|---|
| | Enrolment | Allocation | Post-allocation | | | Close-out |
| TIMEPOINT | $-t_1$ | 0 | $t_1$<br><br>Baseline | $t_2$<br><br>6 months | $t_3$<br><br>12 months | $t_4$ |
| ENROLMENT: | | | | | | |
| Eligibility screen | X | | | | | |
| Informed consent | X | | | | | |
| Randomization | | X | | | | |
| Allocation | | X | | | | |
| INTERVENTIONS: | | | | | | |
| AFB | | | ▬▬▬ | | | |
| EFB | | | ▬▬▬▬▬ | | | |
| SOC | | | ▬▬▬▬▬▬▬▬ | | | |
| ASSESSMENTS: | | | | | | |
| Socio- Demographic | | | X | | | |
| Primary Outcome: Feasibility | | | ▬▬▬▬▬▬ | | | |
| Primary Outcome: Acceptability | | | | X | X | |
| Primary Outcome: Fidelity | | | ▬▬▬▬▬▬ | | | |
| Secondary Outcome: Effectiveness | | | X | X | X | |

**Fig 1. Study schedule of enrolment, interventions, and assessments.** AFB: Adapted Friendship Bench, EFB: Enhanced Friendship Bench, SOC: Standard of Care.

population. Each of the four sites provides ART services and has a monthly ART clinic volume of more than 115 individuals (age 10–19) who are either initiating care, re-initiating care, or are established patients. We believe these clinical volumes will provide adequate numbers for our recruitment efforts. UNC Project has previously successfully recruited and enrolled adolescents in multiple longitudinal studies at these sites.

## Participants and sample size

**Participants.** Adolescents will be recruited in this study if they meet the eligible criteria as follows: (1) aged 13–19; (2) diagnosed with HIV (vertically or horizontally); (3) scored ≥ 13 on the self-reported and previously used BDI-II [36]; (4) living in the clinic's catchment area with intention to remain in their current residence for at least 1 year (duration of study enrollment); and (5) willing to provide written informed consent (adolescents ≥ 18 or 16–17

years old and those married will provide their own consent form; for those from 13–17 years old, consent form will be made with their legal representatives). The BDI-II consists of 21 items with a two week recall period. Study staff will administer the BDI-II during the screening process. BDI-II scores are classified as minimal (0–13), mild (14–19), moderate (20–28), and severe (29–63). In a validation study amongst ALWH in Malawi, a BDI-II score of ≥ 13 achieved sensitivity of more than 80% [36].

**Sample size considerations.** We will enroll 105 total participants for this randomized pilot study (n = 35 ALWHs for each arm) with approximately 27 participants at each of the four study sites. Given participants will be randomized, we expect balance across clinics. The enrollment will be performed for 8 weeks, we expect to recruit 3 participants per week in each clinic that ensures equal tentative numbers of enrollments for the four clinics. As we explore feasibility, acceptability and fidelity of the interventions from a pilot study lenses, we do not calculate sample size for our main outcomes. The sample of 105 ALWH (35 per arm) will be sufficient to estimate quantitative measures of feasibility and acceptability with reasonable precision (e.g., confidence intervals around proportions of ± 7–9 percentage points across all arms, and ± 9–17 percentage points within a given arm).

## Intervention description

The original protocol of the Friendship Bench consists of six individual counseling sessions plus optional group peer support [31]. In the prior phase of the study, we conducted formative research (in-depth interviews, focus groups, and social support mapping sessions) with ALWH, health care providers, caregivers, and previous young participants and implementors of a FB intervention for perinatal women living with HIV and depression, to examine ALWHs' experiences of depression, its impact on HIV care engagement, and intervention preferences. Drawing from formative work, we adapted the original Friendship Bench intervention protocol for ALWH to meet their developmental and contextual needs and further enhanced the intervention with peer support to facilitate engagement in HIV care [37,38]. In this study protocol, we describe a pilot RCT phase where we evaluate the adapted and enhanced FB.

The AFB will include 6 individual counseling sessions facilitated by counselors attached to one of the study clinics (Table 1). We will conduct one counseling session every 7–10 days, instead of one session per week as in the original FB. The first session will include three components called Opening the Mind, Uplifting, and Strengthening, and the subsequent sessions will be built on the first. Throughout the first session, participants are encouraged to open their minds to identify their problems, choose one to work on, identify a feasible solution, and agree on an action plan through an iterative process guided by the counselor. To be youth-friendly adapted and address the unique needs of ALWH, young counselors (aged 18–21 years old) who are motivated in working with adolescents will be selected and trained on relevant topics such as depression, adolescent development, youth-friendly counseling skills, problem-solving therapy,

Table 1. Key features of intervention arms.

| Elements | Delivered by | Standard of care | Adapted FB | Enhanced FB |
|---|---|---|---|---|
| Basic mental health care and referral for additional care (if indicated, for severe cases) | Staff in public facilities in the study clinics | ✓ | ✓ | ✓ |
| | Trained study nurses | ✓ | ✓ | ✓ |
| Six counseling sessions (every 7–10 days) | Trained counselors | | ✓ | ✓ |
| Six peer support sessions (monthly) | Peer supporters | | | ✓ |

HIV care and self-care. Counseling will be available outside of school time and on weekends to ensure accessibility. Each session will last 30 to 45 minutes and be conducted in a private clinic room in the participant's local language (Chichewa). Any participant that indicates any degree of suicidal ideation will be referred for further in-person assessment of their suicide risk using the Suicide Risk Assessment Protocol [39]. Referral for additional care (e.g., case management by a supervisor trained in mental health) will be provided for those who do not improve after 4 sessions of individual therapy or having suicidal ideation.

The EFB will include all elements of the AFB but will additionally integrate peer support to facilitate engagement in care for ALWH with depression. Peer support will include six in-person monthly group sessions with the following topics: (1) mental health and HIV, (2) HIV status communication and disclosure, (3) ART adherence and viral load testing, (4) sex/relationships and secondary prevention, (5) stigma, and (6) planning for the future. As suggested from aforementioned formative work, one group session per month is ideal for peer support component. Along with trained counselors as described in the AFB, the EFB will also involve trained, youth-friendly peer supporters living with HIV who are aged 18–21 years old and motivated to work with ALWH.

SOC for mental health in public facilities in Malawi includes options for basic supportive counseling, medication, referral to the mental health clinic, or referral to the psychiatric units at tertiary care hospitals for more severe cases. In this study, SOC will be enhanced by a trained study nurse who will provide mental health evaluation; brief supportive counseling; information, education, and support on depression; and facilitation of referral to the appropriate clinics/hospitals if indicated.

## Primary and secondary outcomes

The primary outcomes of this pilot study are feasibility, acceptability, and fidelity of the adapted and enhanced Friendship Bench interventions in reducing depressive symptoms and improving engagement in HIV care among ALWH (Table 2). These outcomes will be assessed at 6 months and 12 months and compared across 3 arms.

Feasibility will be defined as the ability to successfully enroll and retain depressed ALWH in the study. Feasibility will be assessed as the number of ALWH enrolled, a comparison of planned to actual enrollment and reasons for non-enrollment, and the proportion of ALWH retained in each arm by the end of the study period. We will also assess the number of sessions that participants attend out of the total number of sessions offered.

Fidelity will be defined as adherence to the intervention protocol. Fidelity to content for sessions will be reviewed using audio recording for the sessions by a member of the research

**Table 2. Descriptions of outcomes and indicators in the study.**

| Outcome | Indicator | Source | Time point |
|---|---|---|---|
| Feasibility | • # ALWH enrolled compared to the expected number <br> • Reason for non-enrollment <br> • % ALWH retained in each arm | Study visit attendance form, counseling session attendance form, peer support engagement logs | Throughout study period |
| Fidelity | • ≥ 80% content in checklist covered | Counseling session checklist | Throughout study period |
| Acceptability | • Experiences of participants and study staff about the interventions <br> • Experiences of participants with counseling sessions, peer support engagement | Qualitative data using in-depth interviews (Brief exit interviews); Survey structured questionnaire | At 6 months and 12 months |
| Preliminary effectiveness | • Changes in BDI-II score <br> • % ALWH attended at least one appointment per quarter <br> • % ALWH with HIV RNA < 1000 copies/mL | Survey structured questionnaire | At baseline, 6 months, and 12 months |

team using a checklist. The checklist consists of 10 items corresponding to 10 core session components rating from "1 – Component not done" to "4 – Component exceeds expectations" for each component. Covering at least 80% of checklist items will be considered fidelity to the intervention protocol [40].

Acceptability will be defined as the ability to deliver a culturally and resource-appropriate intervention. Acceptability will be assessed through both survey structured questionnaire and brief exit interviews/in-depth interviews. Brief exit interviews will be conducted with all HIV providers treating patients in the FB protocol, professional counselors, peer supporters, and supervising Master Trainers; and a subset of enrolled ALWH. For ALWH, exit interviews will occur after the final FB session (approximately 6–9 weeks after beginning intervention), while exit interviews with other groups occur after the end of FB activities at each site. Exit interviews included both closed and open-ended questions exploring participants' satisfaction level, how easy the intervention was to participate in or deliver, the perceived usefulness of the intervention, suggestions for improvement, and will explore contextual factors that impeded or facilitated implementation. We aim to include 10 enrolled ALWH per arm (out of 35 per arm) with respect to the rule of thumb to achieve saturation in these main themes. We conveniently sample ALWH based on their socio-demographic characteristics to make sure of the diversity of the qualitative sample.

Effectiveness will be considered as a secondary outcome and will be assessed through changes in the three following indicators: 1) depressive symptoms (≥50% change in BDI-II scores from enrollment to the 6 months and 12 months), 2) retention in care (whether an ALWH attended at least one appointment per quarter), 3) and viral suppression at 6 months and 12 months (HIV RNA < 1000 copies/mL).

## Data collection

In terms of the quantitative data for primary outcomes, study staff will collect programmatic data related to enrollment, retention, counseling session attendance, adherence to intervention protocols during study visits, counseling sessions, peer support session attendance, and participant tracing. Quantitative data will be collected and managed using Research Electronic Data Capture (REDCap). Exit interviews will be digitally recorded, transcribed in Chichewa, and then translated into English by study staff according to a transcription protocol. All transcripts will then be reviewed by the interviewer for transcription and translation accuracy.

Study staff will also administer a survey to participants at enrollment, 6- and 12-month study visits. The survey questionnaire will include information related to socio-demographic characteristics, social support, mental health, HIV care and stigma, sexual partnership, physical and sexual violence, substance use, and delivery of the Friendship Bench intervention (i.e., experiences with counseling sessions, peer support engagement). Viral load data will also be collected at enrollment, 6- and 12-month study visits.

## Statistical analysis

We will compare characteristics of participants at enrollment to assess balance across 3 study arms with t-tests for continuous variables and chi-square tests for categorical variables. Quantitative measures of feasibility, fidelity, acceptability, and preliminary effectiveness will be summarized using means and standard deviations or proportions and be compared across 3 study arms as appropriate. Changes in depressive symptoms and engagement in HIV care will be assessed at the 6-month and the 12-month study visits in comparison with the enrollment visit. Intent-to-treat (ITT) and per-protocol analyses will be considered to compare across 3 study arms if appropriate. Missing data will be reported using count and multiple imputation

will be performed if needed. Analysis of qualitative exit in-depth interview data to assess acceptability compartment will involve four steps: 1) reading for content; 2) deductive and inductive coding; 3) data display to identify emerging themes; and 4) interpretation. Codes will be refined during the analysis process and memos will be written for each identified theme.

## Ethics and confidentiality

The Institutional Review Boards of the University of North Carolina at Chapel Hill approved this study (UNCPM 22319, date of approval 2/16/2024). All data will be kept in a locked cabinet at UNC project or on a secure server at UNC Project accessible only to approved study personnel. All eligible and interested participants aged 18 and above, or legally emancipated 16–17-year-old minors who were married, will be asked for written informed consent. All eligible and interested participants aged 13–17 will be asked to provide written assent with written parental consent. Before study enrollment, research staff will engage participants in a consent comprehension activity where they will ask participants a series of questions to ensure understanding of the study. All participants (and guardian if present) will receive travel reimbursement for attending study visits, counseling sessions, and peer support groups.

## Discussion

ALWH face a number of unique challenges during the transition to adulthood including inconsistent social support, safe sex negotiation, status disclosure to their social ties, HIV stigma, and barriers in accessing health services [3] – all of which may increase the risk of depression, low HIV care engagement and poor HIV treatment outcomes. While ALWH in SSA are considerably affected by the dual burden of HIV and mental health conditions, there are limited interventions primarily addressing depression and HIV care engagement for this population. Indeed, providing accessible, confidential, and developmentally appropriate care or a safe space where adolescents can share sensitive concerns without judgement is critically crucial. The Friendship Bench intervention has demonstrated effectiveness in improving depression in adults [30], and is being adapted for ALWH in Zimbabwe but have yet to incorporate peer support to enhance HIV care engagement. In the prior phase of the current pilot, we conducted formative work to adapt the Friendship Bench model to be youth friendly for ALWH in Malawi and enhance it with group peer support for HIV care engagement. Preliminary results from our formative work suggest it is critical to integrate counseling and peer support to address both depression and HIV care engagement for ALWH in Malawi [38]. We found that experienced respectful young counselors, time flexibility and confidentiality in location and counselor are crucial for a youth-friendly adapted model [41]. Likewise, confidentiality and trust appear to be key for peer support component. Peers with HIV and lived experience, peer support with group setting are more preferred [41].

The current pilot randomized controlled trial will allow for evaluating whether the adapted and enhanced FB models are appropriate and effective for ALWH in Malawi. First, we conducted formative work to adapt and enhance existing FB protocols to align with the specific needs and characteristics of ALWH in Malawi. This clinical trial is to pilot the adapted FB we developed and integrated peer support to enhance HIV care engagement for this population. Second, this pilot will contribute to the body of evidence for the potential feasibility, fidelity, acceptability, and preliminary effectiveness of the AFB and EFB and will inform the potential of scale up of these interventions for ALWH in Malawi. To our knowledge, there are few similar published assessments beyond Youth FB for ALWH in Zimbabwe around 2018–2020 [33,42]. Third, this pilot will provide insight into the resource needs and the uptake of

new-established FB models that can lay the groundwork for a longer-term plan to assess effectiveness in a stronger-powered RCT, but also could help modify the FB models to better align with real world situations.

The study may be subject to several potential limitations. First, as the adapted and enhanced FB models are novel in Malawi, this pilot may face challenges in controlling the consistency of the intervention delivery over three arms and four study clinics. By thoroughly training lay workers and peer supporters to conduct the interventions and by building systematic reporting/managing systems, we expect the intervention consistency will be met. Second, the feasibility outcome of the interventions (i.e., retention in AFB and EFB) may be overestimated due to the influence of other co-occurring activities that require participants to come to clinics. Well-documenting co-occurring activities will help inform proper analysis strategies to reduce this bias (e.g., stratification). Third, participants in the same clinics may be assigned in different intervention arms as a result of individually randomized, there is a potential risk of contamination across intervention arms. Efforts will be made to prevent contamination and interaction amongst participants in different arms attending the same clinic.

## Conclusion

Resource-appropriate interventions to address depression and HIV care engagement among ALWH in Malawi are urgently needed. This pilot clinical trial will provide insights into youth-friendly Friendship Bench models for ALWH in Malawi and the value of adding group peer support for HIV care engagement. The information gathered in this study will lead to a R01 application to test our adapted intervention in a large-scale cluster randomized controlled trial to improve depression and engagement in HIV care among ALWH.

## Supporting information

**S1 Table. SPIRIT 2013 Checklist: Recommended items to address in a clinical trial protocol and related documents\*** .
(DOCX)

**S1 File. Study protocol (HEADS-UP).**
(PDF)

## Author contributions

**Conceptualization:** Bradley N. Gaynes, Brian W. Pence, Nivedita L. Bhushan.

**Data curation:** Steven M. Mphonda.

**Investigation:** Steven M. Mphonda.

**Methodology:** Bradley N. Gaynes, Brian W. Pence, Nivedita L. Bhushan.

**Project administration:** Hillary Mortensen.

**Resources:** Nivedita L. Bhushan.

**Supervision:** Bradley N. Gaynes, Nivedita L. Bhushan.

**Validation:** Bradley N. Gaynes, Nivedita L. Bhushan.

**Writing – original draft:** Thuy Thi Dieu Dao.

**Writing – review & editing:** Thuy Thi Dieu Dao, Bradley N. Gaynes, Brian W. Pence, Steven M. Mphonda, Kazione Kulisewa, Michael Udedi, Melissa A. Stockton, Jack Kramer, Katherine Grace Waddell, Maria Faidas, Hillary Mortensen, Nivedita L. Bhushan.

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
