## [Decision Letter · Decision Letter 0]

11 Jul 2024

PONE-D-24-12487Friendship Bench Intervention to Address Depression and Improve HIV Care Engagement Among Adolescents Living with HIV in Malawi: Study Protocol for a Pilot Randomized Controlled TrialPLOS ONE

Dear Dr. Dao,

Thank you for submitting your manuscript to PLOS ONE. After careful consideration, we feel that it has merit but does not fully meet PLOS ONE’s publication criteria as it currently stands. Therefore, we invite you to submit a revised version of the manuscript that addresses the points raised during the review process.

Major revisions to the methods sectionMajor revisions to the data analysis sectionMajor revisions to the results section

Please see the feedback from 3 reviewers and kindly address these queries before re-submission. 

We look forward to receiving your revised manuscript.

Kind regards,

Candice Maylene Chetty-Makkan, MA, PhD

Academic Editor

PLOS ONE

Journal Requirements:

3. We note you have included a table to which you do not refer in the text of your manuscript. Please ensure that you refer to Tables 1 and 2 in your text; if accepted, production will need this reference to link the reader to the Table.

Reviewers' comments:

Reviewer's Responses to Questions

**Comments to the Author**

1. Does the manuscript provide a valid rationale for the proposed study, with clearly identified and justified research questions?

Reviewer #1: Yes

Reviewer #2: Yes

Reviewer #3: Partly

2. Is the protocol technically sound and planned in a manner that will lead to a meaningful outcome and allow testing the stated hypotheses?

Reviewer #1: Yes

Reviewer #2: Yes

Reviewer #3: No

3. Is the methodology feasible and described in sufficient detail to allow the work to be replicable?

Reviewer #1: Yes

Reviewer #2: Yes

Reviewer #3: No

4. Have the authors described where all data underlying the findings will be made available when the study is complete?

Reviewer #1: No

Reviewer #2: No

Reviewer #3: No

5. Is the manuscript presented in an intelligible fashion and written in standard English?

Reviewer #1: Yes

Reviewer #2: Yes

Reviewer #3: Yes

6. Review Comments to the Author

You may also provide optional suggestions and comments to authors that they might find helpful in planning their study.

Reviewer #1: Thank you to the authors for the opportunity to review the article. The researchers provide a detailed and well-structured protocol for a pilot RCT aimed at addressing the mental health and HIV care management needs of ALWH in Malawi through the Friendship Bench intervention. The article successfully lays the groundwork for future research to assess the effectiveness of the proposed intervention in a large-scale RCT.

The article is comprehensive and well-written. I provide a handful of comments to improve the protocol.

Introduction

The FB is briefly mentioned in the introduction, but it’s not clear what the intervention is. A sentence or two after the first mention is necessary.

Methods

Page 5, line 6: “In the prior phase of the study,” – There has been no mentioned of a “prior phase of the study”. Any earlier piloting of the intervention/study by the researchers should be mentioned in the methodology section. It needs to be made clear what formative work has been done/you plan to do to adapt the FB intervention to the local setting.

Typos:

The following sentence is confusing and a little too long.

13 While pharmacological treatments are challenging due to cost, availability of designated professionals 14 and mainly reserved for cases of clinically diagnosed, moderate to severe depression, psychological 15 interventions (known as psychotherapy) are preferable and widely implemented for adolescents and 16 young adults with or without HIV in low- and middle-income countries (Bhana et al., 2020, 2021; 17 Sequeira et al., 2022).

There seems to be a word missing after “not been”.

Page 4, line 4: It is currently being adapted for adolescents but has not been specifically for ALWH in Botswana

5 (Brooks et al., 2019).

Considerations:

The one major point that is lacking from the protocol is considerations around scaling the intervention. The one barrier that comes to mind is the availability of healthcare workers to provide the intervention outside a research context. Also, will the peer component be funded, in which case, is there scope for this to be a paid mandate in Malawi. How has the model scaled among adults in other countries?

Reviewer #2: The protocol paper has clearly defined the rationale, methods, intervention and data collection procedures. The pilot study addresses a very important issue of mental health among ALWH; using an advanced and enhanced version of the Friendship bench intervention. It will be interesting to see results from this pilot and how they influence further adaptation of the intervention and a large-scale RCT. Well done to the authors.

Please see some comments to address below:

1. Page 3: line 7-9 Indeed, comorbid depression is prevalent and burdensome amongst ALWH in SSA and depression is a significant threat to engagement and retention in HIV care as well as worsening HIV related outcomes among those receiving HIV treatment”

Comment: Are there any stats from past or recent studies showing the burden of depression among adolescents and more specifically among ALWH or HIV care disengagement rates among ALWH with depression or mental health challenges?

2. Page 4: line 4 It is currently being adapted for adolescents but has not been specifically for ALWH in Botswana.

Comment: The sentence is unclear, reference to Botswana is a bit confusing. Please rephrase.

3. Page 4-5: Study settings and participants & sample size

Comment: What is the justification for selecting 4 clinics? Please also a clarify if by patient volume you men these are high volume clinics? Are these public health clinics?

Please expand briefly on the Beck’s Depression Inventory: what is it and what does a score of ≥ 13 mean. Please add a justification for including those who scored ≥ 13 and why these are an important sub-group to include? Will the BDI-II be completed before enrolment or is this coming from a different data source- if this was previously completed, what was the cut-off date i.e. The score is valid for how long after the test was taken?

Please clarify eligibility criteria #4 “living in the clinic’s catchment area with intention to remain for at least 1 year” – what does “intention to remain” mean here? Does it refer to their place of residence within the clinic’s catchment area or remain receiving services from the clinic they were enrolled in?

What is the expected enrolment distribution per clinic and arm? How will you ensure that the enrolments are not skewed towards 1 clinic and that the study arms are not skewed by clinic enrolments? Are there any other socio-economic or structural differences across the 4 clinics that will need to be considered and how they could potentially influence the pilot results?

4. Page 5: line 1 & 7

Comment: Please correct the abbreviation, it is written as “AWLH” instead of ALWH – please also check the whole manuscript, there are a few more similar errors.

5. Page 5: Intervention description

Line 10 – Please add that AFB will be “individual” counselling sessions

Line 10-11 – remove reference to “HIV clinics” – revise to refer to these as study clinics – in essence these are still clinics providing general health services with integrated ART?

Line 37 – The frequency of the support session: please add a justification for holding monthly peer support sessions if the individual sessions are every 7-10 days.

General comments: in the delivery of the peer support intervention, are the younger adolescents (13-14/15 separated from the older adolescents (16-19) also separated by gender? How do you ensure inclusivity and maximum participation in the groups, the topics may be the same but engagement for younger and older adolescents and males may differ, being combined in the same session may lead to minimal participation from other age groups and gender?

Reviewer #3: General comments:

This protocol manuscript describes an intended pilot trial of a peer support intervention for adolescents living with HIV (ALWH).

I have a few major concerns about this manuscript. There are many details about the analyses, study design, and procedures that I feel are needed.

Most importantly, there are no sample size calculations in this manuscript which must be included per PLOS ONE guidelines https://journals.plos.org/plosone/s/submission-guidelines#loc-study-protocols. This applies both on p.5 to the enrollment numbers but also to the subset that receive the qualitative evaluation.

In addition, I didn't realize there was a qualitative outcome until the data analysis section. There needs to be more detail on how these analyses will be performed. Listing out the steps gives a general idea, but I don't know how will coding be performed, or what sort of methods will be used.

I also could not locate a data management plan or a section on safety considerations in the manuscript.

Regarding the study design and analysis methods, one comment I have is, since the intervention was delivered in small groups, I believe this is an individual randomized group treatment trial (IRGT doi: 10.2105/AJPH.2007.127027). In IRGTs, the composition of the groups could influence the participant's outcome; in other words, within group responses may be correlated. If this is true, then group membership must be taken into account meaning the power will be lower than in an individually-randomized trial.

Specific comments:

1. (p.4, lines 21-22) Which software will you use? The random number generators are not all created equal.

2. (p.4, lines 23-24) I can see that blinding might not be possible, but I think this is too vague for a protocol.

3. (p.5, lines 1-3) No sample size calculations? See general comments.

4. (p.6, line 17) How will you be sampling people for the exit interviews? Also, if these interviews are brief, why are they not performed with all people?

5. (p.7, line 6) At no point before this is it mentioned that acceptability will be assessed qualitatively. This should be made clear in the methods section. I would also encourage you to split the outcomes into quantitative and qualitative as well and to be more clear about the qualitative methods that are to be used in this study. I also wonder whether 10 participants in each arm is sufficient for qualitative analysis.

6. (p.7) How will data be collected? Paper and pencil? ODK? A web-based system? How will data recording errors be minimized?

7. (p.7, lines 18-21) I strongly encourage the authors to use standardized mean differences instead of p-values to compare baseline characteristics, especially in a small trial like this. Significance testing in these situations is generally frowned upon because a non-significant p-value does not indicate that groups are the same. For info on the topic in relation to baseline imbalance in randomized trials see Altman, https://doi.org/10.2307/2987510 and Senn, https://doi.org/10.1002/sim.4780131703. My recommendation is to use standardized difference to assess differences (see Austin, https://doi.org/10.1080/03610910902859574).

7. PLOS authors have the option to publish the peer review history of their article (what does this mean? ). If published, this will include your full peer review and any attached files.

**Do you want your identity to be public for this peer review?** For information about this choice, including consent withdrawal, please see our Privacy Policy .

Reviewer #1: No

Reviewer #2: No

Reviewer #3: No

---

## [Author Response · Author response to Decision Letter 0]

20 Aug 2024

Reviewer 1

1. FB is briefly mentioned in the introduction, but it’s not clear what the intervention is. A sentence or two after the first mention is necessary.

Response: We thank the reviewer for this comment. We added one sentence after the first mention of FB in introduction:

The original Friendship Bench consists of six individual counseling sessions plus optional group peer support.

2. Page 5, line 6: “In the prior phase of the study,” – There has been no mentioned of a “prior phase of the study”. Any earlier piloting of the intervention/study by the researchers should be mentioned in the methodology section. It needs to be made clear what formative work has been done/you plan to do to adapt the FB intervention to the local setting.

Response: We thank the reviewer for raising this point. We used the ADAPT-ITT framework and this pilot RCT serves as step 7-8 in the framework. We added more information in text:

In the prior phase of the study, we conducted formative research (in-depth interviews, focus groups, and social support mapping sessions) with ALWH, health care providers, caregivers, and previous young participants and implementors of a FB intervention for perinatal women living with HIV and depression, to examine ALWHs’ experiences of depression, its impact on HIV care engagement, and intervention preferences. Drawing from formative work, we adapted the original Friendship Bench intervention protocol for ALWH to meet their developmental and contextual needs and further enhanced the intervention with peer support to facilitate engagement in HIV care [37,38]. In this study protocol, we describe a pilot RCT phase where we evaluate the adapted and enhanced FB.

3. The following sentence is confusing and a little too long.

While pharmacological treatments are challenging due to cost, availability of designated professionals 14 and mainly reserved for cases of clinically diagnosed, moderate to severe depression, psychological 15 interventions (known as psychotherapy) are preferable and widely implemented for adolescents and 16 young adults with or without HIV in low- and middle-income countries (Bhana et al., 2020, 2021; 17 Sequeira et al., 2022).

Response: We thank the reviewer for this comment. We edited the sentence as below:

Pharmacological treatments are challenging due to cost and availability of designated professionals and are often reserved for treating moderate to severe cases of depression. In contrast, psychological interventions are preferable and widely implemented for adolescents and young adults with or without HIV in resource-limited countries [10–12].

4. There seems to be a word missing after “not been”.

Page 4, line 4: It is currently being adapted for adolescents but has not been specifically for ALWH in Botswana (Brooks et al., 2019).

Response: We thank the reviewer for this comment. We edited the sentence as below:

It is currently being adapted for adolescents in Botswana but has not been specifically adapted for ALWH.

Considerations:

The one major point that is lacking from the protocol is considerations around scaling the intervention. The one barrier that comes to mind is the availability of healthcare workers to provide the intervention outside a research context. Also, will the peer component be funded, in which case, is there scope for this to be a paid mandate in Malawi. How has the model scaled among adults in other countries?

Response: We thank the reviewer for raising this important point. The availability of lay workers and peer supporters to implement the intervention outside of a research context is indeed critical for future scale-up. Our current pilot work is focused on evaluating the feasibility, acceptability, fidelity, and preliminary effectiveness of implementing this evidence-based intervention for ALWH in Malawi. We plan to explore facilitators and barriers to scale-up during out exit interviews with study staff, clinicians, and Malawi Ministry of Health officials.

Reviewer 2

1. Page 3: line 7-9 Indeed, comorbid depression is prevalent and burdensome amongst ALWH in SSA and depression is a significant threat to engagement and retention in HIV care as well as worsening HIV related outcomes among those receiving HIV treatment”

Comment: Are there any stats from past or recent studies showing the burden of depression among adolescents and more specifically among ALWH or HIV care disengagement rates among ALWH with depression or mental health challenges?

Response: We thank the reviewer for this comment. We added some descriptive statistics for the prevalence of depressive symptoms among ALWH in SSA.

Indeed, comorbid depression is prevalent and burdensome amongst ALWH in SSA with estimates of depressive symptoms among ALWH ranging from 12% to 45% [4].

2. Page 4: line 4 It is currently being adapted for adolescents but has not been specifically for ALWH in Botswana.

Comment: The sentence is unclear, reference to Botswana is a bit confusing. Please rephrase.

Response: We thank the reviewer for this comment. We edited the sentence as below (also, see response # 4 above to the first reviewer):

It is currently being adapted for adolescents in Botswana but has not been specifically adapted for ALWH.

3. Page 4-5: Study settings and participants & sample size

Comment: What is the justification for selecting 4 clinics? Please also a clarify if by patient volume you men these are high volume clinics? Are these public health clinics?

Please expand briefly on the Beck’s Depression Inventory: what is it and what does a score of ≥ 13 mean. Please add a justification for including those who scored ≥ 13 and why these are an important sub-group to include? Will the BDI-II be completed before enrolment or is this coming from a different data source- if this was previously completed, what was the cut-off date i.e. The score is valid for how long after the test was taken?

Please clarify eligibility criteria #4 “living in the clinic’s catchment area with intention to remain for at least 1 year” – what does “intention to remain” mean here? Does it refer to their place of residence within the clinic’s catchment area or remain receiving services from the clinic they were enrolled in?

What is the expected enrolment distribution per clinic and arm? How will you ensure that the enrolments are not skewed towards 1 clinic and that the study arms are not skewed by clinic enrolments? Are there any other socio-economic or structural differences across the 4 clinics that will need to be considered and how they could potentially influence the pilot results?

Response: We thank the reviewer for raising these important points. We clarify these points below and add them to the text as appropriate.

Justification for selecting 4 clinics:

Each of the four sites provide ART services and have a monthly ART clinic volume of more than 115 individuals (age 10-19) who are either initiating care, re-initiating care, or are established patients. We believe these clinic volumes will provide adequate numbers for our recruitment efforts. UNC Project has previously successfully recruited and enrolled adolescents in multiple longitudinal studies at these sites.

BDI-II:

The BDI-II consists of 21 items with a two-week recall period. Study staff will administer the BDI-II during the screening process. BDI-II scores are classified as minimal (0-13), mild (14-19), moderate (20-28), and severe (29-63). In a validation study amongst ALWH in Malawi, a BDI-II score of ≥13 achieved sensitivity of more than 80% [36].

Intention to remain:

(4) living in the clinic’s catchment area with intention to remain in their current residence place for at least 1 year (duration of study enrollment).

Distribution per clinic and arm:

We expect to enroll 105 total participants for this randomized pilot study (n=35 ALWH for each arm) with approximately 27 participants at each of the four study sites. Given participants will be randomized, we expect balance across arms, regardless of clinic.

4. Page 5: line 1 & 7

Comment: Please correct the abbreviation, it is written as “AWLH” instead of ALWH – please also check the whole manuscript, there are a few more similar errors.

Response: We thank the reviewer for the thorough review. We double checked and edited this.

5. Page 5: Intervention description

Line 10 – Please add that AFB will be “individual” counselling sessions

Line 10-11 – remove reference to “HIV clinics” – revise to refer to these as study clinics – in essence these are still clinics providing general health services with integrated ART?

Line 37 – The frequency of the support session: please add a justification for holding monthly peer support sessions if the individual sessions are every 7-10 days.

Response: We thank the reviewer for the thoughtful comment. We added “individual” and removed “HIV” in the sentence:

The AFB will include 6 individual counseling sessions facilitated by counselors attached to one of the study clinics.

Regarding the frequency of the sessions, we added this sentence in the paragraph describing AFB:

We will conduct one counseling session every 7-10 days, instead of one session per week as in the original FB.

For frequency of peer support, we revised:

As suggested from aforementioned formative work, one group session per month is ideal for peer support component.

General comments: in the delivery of the peer support intervention, are the younger adolescents (13-14/15 separated from the older adolescents (16-19) also separated by gender? How do you ensure inclusivity and maximum participation in the groups, the topics may be the same but engagement for younger and older adolescents and males may differ, being combined in the same session may lead to minimal participation from other age groups and gender? We thank the reviewer for this comment. Peer support sessions will be conducted separately for younger (13-15) and older (16-19) ALWH. We decided not to separate the groups by gender given the preferences of ALWH in our formative research.

Reviewer 3

General comments:

Most importantly, there are no sample size calculations in this manuscript which must be included per PLOS ONE guidelines https://journals.plos.org/plosone/s/submission-guidelines#loc-study-protocols. This applies both on p.5 to the enrollment numbers but also to the subset that receive the qualitative evaluation.

In addition, I didn't realize there was a qualitative outcome until the data analysis section. There needs to be more detail on how these analyses will be performed. Listing out the steps gives a general idea, but I don't know how will coding be performed, or what sort of methods will be used.

I also could not locate a data management plan or a section on safety considerations in the manuscript.

Regarding the study design and analysis methods, one comment I have is, since the intervention was delivered in small groups, I believe this is an individual randomized group treatment trial (IRGT doi: 10.2105/AJPH.2007.127027). In IRGTs, the composition of the groups could influence the participant's outcome; in other words, within group responses may be correlated. If this is true, then group membership must be taken into account meaning the power will be lower than in an individually-randomized trial.

Response: We thank the reviewer for the review and comments.

Regarding sample size calculation for our qualitative outcome, please see our below response (#3).

We added one sentence regarding data safety in the Ethnics and Confidentiality section.

Please refer to specific comments for clarification.

1. (p.4, lines 21-22) Which software will you use? The random number generators are not all created equal.

Response: We thank the reviewer for this comment. The randomization list is generated using the rand() function in Excel.

2. (p.4, lines 23-24) I can see that blinding might not be possible, but I think this is too vague for a protocol.

Response: We thank the reviewer for this comment. As the nature of counseling intervention, participants might or might not be aware of which type of intervention that they are receiving. Given that, we are trying to enroll participants with consistent contents of informed consent regardless of their assigned arm. Conservatively speaking, we suppose blinding might not be possible.

3. (p.5, lines 1-3) No sample size calculations? See general comments.

Response: We thank the reviewer for this comment. The main goal of this pilot RCT is to evaluate the feasibility, fidelity and acceptability of the interventions. Our secondary outcome is preliminary effectiveness. We are confident that the current sample size is enough to achieve this main goal and sample size calculation will be applied in a larger-scale RCT in the future.

4. (p.6, line 17) How will you be sampling people for the exit interviews? Also, if these interviews are brief, why are they not performed with all people?

Response: We thank the reviewer for raising this question. We will select conveniently a subset of enrolled participants for exit interviews and all providers (incl. counselors, peer supporter, clinical staff). We select participants based on their socio-demographic characteristics to make sure of the diversity of the qualitative sample. We aim to include 10 per arm (i.e., 30 in total). Given the rule of thumb of 5-10 in-depth interviews per stratum to achieve 80% of themes, we expect to achieve saturation within our proposed sample sizes.

We added this information in text (Page 7, in the paragraph of Acceptability outcome) and remove one sentence about the number of participants in qualitative part in Data collection section as we think it is more appropriate to move this part into paragraph explaining the outcome.

5. (p.7, line 6) At no point before this is it mentioned that acceptability will be assessed qualitatively. This should be made clear in the methods section. I would also encourage you to split the outcomes into quantitative and qualitative as well and to be more clear about the qualitative methods that are to be used in this study. I also wonder whether 10 participants in each arm is sufficient for qualitative analysis.

Response: We thank the reviewer for this comment. We first mention the qualitative piece in the “Primary and secondary outcomes” section. As we only use qualitative part as a small piece supported for one of the four outcomes in our study (i.e., acceptability), we seem to prefer keeping it embedded in the acceptability outcome description.

6. (p.7) How will data be collected? Paper and pencil? ODK? A web-based system? How will data recording errors be minimized?

Response: We thank the reviewer for the comment. Quantitative data will be collected and managed using Research Electronic Data Capture (REDCap). We added this information in the data collection section.

7. (p.7, lines 18-21) I strongly encourage the authors to use standardized mean differences instead of p-values to compare baseline characteristics, especially in a small trial like this. Significance testing in these situations is generally frowned upon because a non-significant p-value does not indicate that groups are the same. For info on the topic in relation to baseline imbalance in randomized trials see Altman, https://doi.org/10.2307/2987510 and Senn, https://doi.org/10.1002/sim.4780131703. My recommendation is to use standardized difference to assess differences (see Austin, https://doi.org/10.1080/03610910902859574). Response: We thank the reviewer for this comment. As clarified earlier, the current pilot aims to characterize preliminary descriptions of the interventions. Comparing absolute count/numbers, proportions or continuous values might provide some sense of which parts could be of interest in adhoc analyses or future studies. That said, statistical tests or inferential considerations are less likely to be priorities of this pilot. Also, we are planning to use p values as it is probably intuitive for audiences.

---

## [Decision Letter · Decision Letter 1]

21 Nov 2024

PONE-D-24-12487R1Friendship Bench Intervention to Address Depression and Improve HIV Care Engagement Among Adolescents Living with HIV in Malawi: Study Protocol for a Pilot Randomized Controlled TrialPLOS ONE

Dear Dr. Dao,

Thank you for submitting your manuscript to PLOS ONE. After careful consideration, we feel that it has merit but does not fully meet PLOS ONE’s publication criteria as it currently stands. Therefore, we invite you to submit a revised version of the manuscript that addresses the points raised during the review process.

I reviewed the feedback from the 3 reviewers and you have definitely improved the content of the manuscript by working through the suggestions. Although you have addressed most of the major comments, the statistician who is reviewing this manuscript still has major concerns with lack of detail on the sample size calculation. Please be sure to:

Include more details on the sample size calculationTwo reviewers agree that the manuscript is ready for acceptance. Reviewer 3 would like additional details and motivation on the sample size calculation. Some guidance has been provided on how to address the sample size calculation query.Once the query on the sample size calculation has been addressed, the manuscript will be ready for acceptance.

We look forward to receiving your revised manuscript.

Kind regards,

Candice Maylene Chetty-Makkan, MA, PhD

Academic Editor

PLOS ONE

Additional Editor Comments:

Thank you for addressing the reviewer comments and majority of the responses were addressed. However, the assigned statistician (Reviewer 3) reviewing this manuscript still has significant concerns with the sample size calculation. Can the authors please address this query? Once this major query on the sample size calculation has been addressed, this manuscript meets the essential requirements for publication.

Reviewers' comments:

Reviewer's Responses to Questions

**Comments to the Author**

1. Does the manuscript provide a valid rationale for the proposed study, with clearly identified and justified research questions?

Reviewer #1: Yes

Reviewer #2: Yes

Reviewer #3: Yes

2. Is the protocol technically sound and planned in a manner that will lead to a meaningful outcome and allow testing the stated hypotheses?

Reviewer #1: Yes

Reviewer #2: Yes

Reviewer #3: Partly

3. Is the methodology feasible and described in sufficient detail to allow the work to be replicable?

Reviewer #1: Yes

Reviewer #2: Yes

Reviewer #3: No

4. Have the authors described where all data underlying the findings will be made available when the study is complete?

Reviewer #1: Yes

Reviewer #2: No

Reviewer #3: No

5. Is the manuscript presented in an intelligible fashion and written in standard English?

Reviewer #1: Yes

Reviewer #2: Yes

Reviewer #3: Yes

6. Review Comments to the Author

You may also provide optional suggestions and comments to authors that they might find helpful in planning their study.

Reviewer #1: Thank you, I am satisfied that the authors have incorporated my comments. Given that I have already reviewed the article, I have no further comments.

Reviewer #2: The authors have made all the revisions suggested in the initial review process and provided sound justification where specific comments were not addressed. The revisions have strengthened the manuscript and it is clear how the protocol can be replicated.

The study rationale is clear and provides evidence-based justifications. The study methods are described in detail and justifications provided for the site selection, sample selection and how the intervention will be delivered. There is no sample size calculation and the study is not powered, which may not be an issue since this is a pilot study and they do mention that these will be considered in a large-scale trial that will follow the this pilot.

This is a protocol paper and no data is presented.

Overall the manuscript is well written and the ideas flow in a logical manner.

Reviewer #3: Thank you for your careful consideration of my comments. I have the following responses:

1. Regardless of the protocol aim or your confidence in your projected sample size, the PLOS ONE publication guidelines state that sample size calculations must be included.

2. Excel is not the best choice for random number generation, but it is passable so long as you use Excel 2010 or later. Previous versions used an algorithm with a short cycle length that had poor performance. Microsoft switched to an algorithm with a greater cycle length in 2010. Further details can be found here: https://link.springer.com/article/10.1007/s00180-014-0482-5. Unfortunately, you cannot set the seed in Excel, which means the numbers are not reproducible, but that should be ok for your purposes.

3. The problems with p-values are well documented (e.g., https://doi.org/10.1080/00031305.2016.1154108; http://www.stat.columbia.edu/~gelman/research/published/asa_pvalues.pdf; https://doi.org/10.1080/02664763.2011.567245; https://doi.org/10.1098/rsos.171085; https://doi.org/10.1007/s10654-016-0149-3) and I would argue that they are decidedly NOT intuitive since so many misinterpretations exist. If inferential considerations are not a priority of the manuscript, then why invite inferential conclusions to be made by including p-values?

7. PLOS authors have the option to publish the peer review history of their article (what does this mean? ). If published, this will include your full peer review and any attached files.

**Do you want your identity to be public for this peer review?** For information about this choice, including consent withdrawal, please see our Privacy Policy .

Reviewer #1: **Yes: ** Laura Rossouw

Reviewer #2: No

Reviewer #3: No

---

## [Author Response · Author response to Decision Letter 1]

22 Nov 2024

Reviewer 3:

1. Regardless of the protocol aim or your confidence in your projected sample size, the PLOS ONE publication guidelines state that sample size calculations must be included.

Response: We thank the reviewer for the comment. We made a subsection under the “Participants and sample size” section for “Sample size considerations”.

In the “Sample size considerations”, we added a justification for our sample size. The paragraph can be found in our revised manuscript and as below.

“As we explore feasibility, acceptability and fidelity of the interventions from a pilot study lenses, we do not calculate sample size for our main outcomes. The sample of 105 ALWH (35 per arm) will be sufficient to estimate quantitative measures of feasibility and acceptability with reasonable precision (e.g., confidence intervals around proportions of ±7-9 percentage points across all arms, and ±9-17 percentage points within a given arm).”

2. Excel is not the best choice for random number generation, but it is passable so long as you use Excel 2010 or later. Previous versions used an algorithm with a short cycle length that had poor performance. Microsoft switched to an algorithm with a greater cycle length in 2010. Further details can be found here: https://link.springer.com/article/10.1007/s00180-014-0482-5. Unfortunately, you cannot set the seed in Excel, which means the numbers are not reproducible, but that should be ok for your purposes.

Response: We thank the reviewer for your insights into randomization process. We specified the version of Excel in the manuscript and as below.

“The study team will generate a random sequence using computer-based software (the rand() function in Microsoft 365 Excel version 2410).”

3. The problems with p-values are well documented (e.g., https://doi.org/10.1080/00031305.2016.1154108; http://www.stat.columbia.edu/~gelman/research/published/asa_pvalues.pdf;https://doi.org/10.1080/02664763.2011.567245; https://doi.org/10.1098/rsos.171085; https://doi.org/10.1007/s10654-016-0149-3) and I would argue that they are decidedly NOT intuitive since so many misinterpretations exist. If inferential considerations are not a priority of the manuscript, then why invite inferential conclusions to be made by including p-values?

Response: We thank the reviewer for your comment. We highly appreciate the suggestion to strengthen our statistical analysis. We will take standardized mean differences into account to compare continuous variables across study arms in the data analysis phase of this study.

---

## [Decision Letter · Decision Letter 2]

22 Jan 2025

Friendship Bench Intervention to Address Depression and Improve HIV Care Engagement Among Adolescents Living with HIV in Malawi: Study Protocol for a Pilot Randomized Controlled Trial

PONE-D-24-12487R2

Dear Dr. Dao,

We’re pleased to inform you that your manuscript has been judged scientifically suitable for publication and will be formally accepted for publication once it meets all outstanding technical requirements.

Kind regards,

Tanya Doherty, PhD

Academic Editor

PLOS ONE

Additional Editor Comments (optional):

Reviewers' comments:

Reviewer's Responses to Questions

**Comments to the Author**

1. Does the manuscript provide a valid rationale for the proposed study, with clearly identified and justified research questions?

Reviewer #1: Yes

Reviewer #3: Yes

2. Is the protocol technically sound and planned in a manner that will lead to a meaningful outcome and allow testing the stated hypotheses?

Reviewer #1: Yes

Reviewer #3: Yes

3. Is the methodology feasible and described in sufficient detail to allow the work to be replicable?

Reviewer #1: Yes

Reviewer #3: Yes

4. Have the authors described where all data underlying the findings will be made available when the study is complete?

Reviewer #1: Yes

Reviewer #3: Yes

5. Is the manuscript presented in an intelligible fashion and written in standard English?

Reviewer #1: Yes

Reviewer #3: Yes

6. Review Comments to the Author

You may also provide optional suggestions and comments to authors that they might find helpful in planning their study.

Reviewer #1: Thank you for the opportunity to review. As previously indicated, I have no further comments to the authors.

Reviewer #3: I have no further comments. Thank you for considering my comments. Best of luck implementing your protocol.

7. PLOS authors have the option to publish the peer review history of their article (what does this mean? ). If published, this will include your full peer review and any attached files.

**Do you want your identity to be public for this peer review?** For information about this choice, including consent withdrawal, please see our Privacy Policy .

Reviewer #1: No

Reviewer #3: No

---

## [Editor Report · Acceptance letter]

PONE-D-24-12487R2

PLOS ONE

Dear Dr. Dao,

I'm pleased to inform you that your manuscript has been deemed suitable for publication in PLOS ONE. Congratulations! Your manuscript is now being handed over to our production team.

Kind regards,

on behalf of

Professor Tanya Doherty

Academic Editor

PLOS ONE